# Clomipramine Induced Oxidative Stress and Morphological Alterations in the Prefrontal Cortex and Limbic System of Neonatal Rats

**DOI:** 10.3390/brainsci15101068

**Published:** 2025-09-30

**Authors:** Norma Angélica Labra-Ruíz, Julieta Griselda Mendoza-Torreblanca, Norma Osnaya-Brizuela, Armando Valenzuela-Peraza, Maribel Ortiz-Herrera, Gerardo Barragán-Mejía, Noemí Cárdenas-Rodríguez, Daniel Santamaría-Del Ángel

**Affiliations:** 1Laboratorio de Neurociencias, Instituto Nacional de Pediatría, Secretaría de Salud, Ciudad de México 04530, Mexico; norma_labra@yahoo.com.mx (N.A.L.-R.); julietamt14@hotmail.com (J.G.M.-T.); osnayanorma@hotmail.com (N.O.-B.); valenzuela.peraza2013@hotmail.com (A.V.-P.); noemicr2001@yahoo.com.mx (N.C.-R.); 2Laboratorio de Bacteriología Experimental, Instituto Nacional de Pediatría, Ciudad de México 04530, Mexico; mortizherrera@hotmail.com (M.O.-H.); mbarraganm@pediatria.gob.mx (G.B.-M.)

**Keywords:** oxidative stress, neurotoxicity, inflammation

## Abstract

Although clomipramine (CLO) is widely used as a serotonin reuptake inhibitor, its subchronic administration during the early stages of brain development leads to depressive-like behaviors in adulthood. High doses of CLO have been linked to mitochondrial impairment and increased reactive oxygen species in cells and adult animals. It is unknown whether subchronic administration of this drug at early ages can induce oxidative stress (OS) in adulthood. The objective of this study was to evaluate OS and cellular damage in the prefrontal cortex and limbic system (hippocampus and amygdala) of rats exposed to CLO neonatally. Methods: Forty male Wistar rats were divided into experimental (EXP) and control (CTRL) groups. The EXP animals received CLO (15 mg/kg, twice daily, subcutaneously, postnatal days 5–35); the CTRL animals received saline. At 55 and 85 days of age, the brains were collected for biochemical assays and histological analysis. Results: Rats exposed to neonatal CLO presented significant reductions in reduced glutathione (GSH) and increases in oxidized glutathione (GSSG) and malondialdehyde in both studied regions, especially on day 85. The GSH/GSSG ratio decreased, indicating persistent OS. Histology revealed neuronal degeneration, pyknotic nuclei, cell shrinkage, and disorganized tissue, which progressed from days 55 to 85. Conclusions: Early exposure to CLO can cause long-lasting neurochemical and structural alterations in the brain regions associated with the regulation of emotions and some behavioral responses that can persist over time and affect behavior in adulthood.

## 1. Introduction

Clomipramine (CLO) is a dibenzoazepine-derived tricyclic antidepressant used as a hydrochloride salt for the treatment of obsessive–compulsive disorder (OCD) and disorders with an OCD component, such as depression, schizophrenia, and Tourette’s syndrome [1,2,3]. The Food and Drug Administration-approved use of CLO is to treat OCD in patients aged 10 years and older; however, the off-label use of CLO includes depression, anxiety, neuropathic pain, chronic pain, panic disorder, insomnia, etc. [3].

The specific mechanism of action of CLO is unclear, as its therapeutic effects often take weeks to appear and may reflect compensatory changes in the central nervous system [2]. However, CLO is known to be a potent serotonin reuptake inhibitor with a stronger affinity for the serotonin transporter (SERT) [1]. CLO is not completely selective, as the main active metabolite, desmethylclomipramine, acts as an inhibitor of the noradrenaline transporter (NET) [1,2,3]. Therefore, CLO increases the concentrations of both serotonergic and noradrenergic neurotransmitters in the synaptic cleft and increases transmission [3,4]. CLO also has some ability to block postsynaptic dopamine receptors [5].

Paradoxical to its antidepressant effect, chronic administration of CLO to neonatal rats produces behaviors that resemble a depressive state in adult animals [6,7]. These behavioral abnormalities include decreased sexual activity [8], reduced aggressive behavior [9], reduced motor hyperactivity in stressful situations [10], increased immobility time in the forced swim test [11], and rapid eye movement sleep abnormalities [12]. Some of these features are similar to those found in human endogenous depression [13]. Furthermore, these changes can be reversed by antidepressant treatment [10,13]. The fact that an antidepressant induces depression is explained by the age at which the drug is administered; that is, when CLO is administered in the early stages of life, it affects the formation of brain circuits, preventing them from functioning properly in adulthood [7].

CLO has additional molecular targets that may differentially alter essential cellular processes. A blockade of sodium channels and N-methyl-D-aspartate receptors might account for its effect on chronic pain, particularly the neuropathic type [1]. However, cardiac sodium channel blockade is the principal mechanism of cardiovascular toxicity caused by tricyclic antidepressants [14]. Additionally, CLO might cause cardiotoxicity through free radical generation and oxidative stress, inducing a significant increase in myocardial lipid peroxidation, a significant decrease in reduced glutathione (GSH), and a significant decrease in the activity of glutathione peroxidase (GPx) and superoxide dismutase (SOD) [15]. In addition, there was a positive correlation between the increase in hydroxyl radical generation and different concentrations of CLO in vitro [15]. Furthermore, CLO is associated with mitochondrial deterioration and the production of reactive oxygen species (ROS) in isolated rat hepatocytes [16]. Additionally, its involvement in apoptosis and cell death mechanisms through increased ROS generation has been observed in human glioma cancer cells treated in vitro [17,18,19]. In adult rats subjected to repeated restraint stress, a high dose of CLO increased lipid peroxidation, free radical levels, and the activity of the antioxidant enzymes SOD and catalase (CAT) in some brain regions [20]. However, in primary fibroblasts from rats with anxiety-related behavior treated with CLO at both low and high doses, no significant alterations in cellular metabolism or oxidative stress (OS) levels were observed [21]. Currently, it is unknown whether subchronic administration of CLO at early ages can induce OS in adulthood. Therefore, the objective of the present study was to evaluate the effects of subchronic administration of CLO in neonatal rats on the levels of GSH, oxidized glutathione (GSSG), and malondialdehyde (MDA) in the limbic system (LIMB; amygdala and hippocampus) and the prefrontal cortex (PCX) to determine the presence of OS and cellular damage at 55 and 85 days of age in these rats.

## 2. Materials and Methods

### 2.1. Animals and Treatment

Forty neonatal male Wistar rats were used, and they were randomly divided into 2 experimental groups (EXP55 and EXP85), each with their respective control groups (CTRL55 and CTRL85). The purpose of analyzing the effect of neonatal subchronic administration of CLO at 55 and 85 days of age was to identify whether the damage caused by the drug was sustained over time. First, the EXP groups received clomipramine hydrochloride (Santa Cruz Biotechnology, Bergheimer Str, Heidelberg, Germany) at 15 mg/kg/12 h subcutaneously from days 5 to 35 of age. The CTRL groups received saline solution (0.9% PiSA, México City, México). Afterwards, to carry out biochemical analyses, on day 55 of age, animals from the EXP55 group (*n* = 8) and animals from the CTRL55 group (*n* = 8) were sacrificed. On day 85 of age, animals from the EXP85 group (*n* = 8) and animals from the CTRL85 group (n = 8) were sacrificed. To carry out the histology study, on day 55 of age, animals from the EXP55 group (*n* = 2) and from the CTRL55 group (*n* = 2) were sacrificed. On day 85 of age, animals from the EXP85 group (*n* = 2) and from the CTRL85 group (*n* = 2) were sacrificed (Figure 1).

Neonatal pups were housed with their mothers in acrylic containers with regulated temperature (22 ± 2 °C) and humidity conditions, a light/dark cycle (12:12 h, lights on at 06:00), and free access to water and food. On day 21 of age, the pups were separated from their mothers and housed in groups of 4 in acrylic boxes. Once the subcutaneous administration was completed (day 35), the animals remained under the conditions described above until sacrifice at day 55 or day 85 (Figure 1). All procedures followed the National Institutes of Health Guide for the Care and Use of Experimental Animals and the Mexican official norm, NOM-062-ZOO-1999. The protocol was approved by the Investigation Committee and by the Institutional Committee for the Care and Use of Laboratory Animals of Instituto Nacional de Pediatría (INP; register number INP-027/2018).

### 2.2. Biochemical Determinations

Animals from the EXP55, CTRL55, EXP85, and CTRL85 groups were sacrificed via decapitation to dissect the LIMB (hippocampus and amygdala) and PCX to assess OS. Since the amygdala boundaries were not clearly defined and the amount of tissue was minimal, the hippocampus and amygdala were obtained together. Protein determination was performed on homogenates from all study groups according to the Lowry method [22]. A portion of approximately 250 mg of tissue was homogenized in 3.75 mL of phosphate-EDTA buffer (Merck, KGaA, Darmstadt, Germany) and 1 mL of 25% H_3_PO_4_ (Merck, KGaA, Darmstadt, Germany). The mixture was subsequently centrifuged at 4 °C at 100,000× *g* for 30 min, and the supernatant was used for the determination of the GSH, GSSG, and MDA contents.

To measure GSH, aliquots of 0.5 mL to 4.5 mL of phosphate buffer (pH 8.0) were added and mixed. One hundred microliters of the mixture were added to 1.8 mL of phosphate buffer (pH 8.0; NaH_2_PO_4_; Na_2_HPO_4_; J. T. Baker, Easton, Pensilvania, USA) and 100 μL of O-phthalaldehyde (OPA; Thermo Fisher Scientific, Madrid, Spain). The new mixtures were incubated at room temperature for 15 min. Their fluorescence signals were recorded at an emission wavelength of 420 nm and an excitation wavelength of 350 nm in a PerkinElmer LS55 spectrofluorometer (Perkin-Elmer Life and Analytical Sciences, Springfield, IL, USA). To measure GSSG, the second fraction of 0.5 mL of the supernatant was added to 200 μL of 0.04 M N-ethyl-maleimide (Sigma-Aldrich, Darmsatdt, Germany) to prevent the oxidation of GSH to GSSG. The mixtures were incubated at room temperature for 30 min, after which 4.3 mL of 0.1 N NaOH (Merck, KGaA, Darmstadt, Germany) was added. One hundred microliters of the resulting mixture were added to 1.8 mL of 0.1 N NaOH plus 100 μL of OPA. The samples were then incubated at room temperature for 15 min, and their fluorescent signals were recorded under the same detection conditions as those for GSH. The results are expressed as nmol/mg protein of GSH or GSSG or as the GSH/GSSG ratio [23].

To determine lipid peroxidation, substances that are reactive to thiobarbituric acid (TBARS; Merck, KGaA, Darmstadt, Germany) were assayed. It was carried out in homogenates of LIMB and PCX to evaluate the amount of MDA produced by free radical damage to cellular components, as described by Gutteridge et al. [24]. Two milliliters of working solution were added to 1 mL of homogenate and incubated in a Thermomix 1420 bath (B. Braun, Melsugen, Germany) at boiling temperature for 30 min. The samples were subsequently placed on ice for 5 min and centrifuged at 100.7× *g* for 15 min in an Orvall RC-5B DuPont centrifuge (Dupont, Bellport, NY, USA). The absorbances of the supernatants were read at 532 nm in a UNICAM Helios-K spectrophotometer (Sistemas Analíticos, UNICAM, Lisboa, Portugal) and interpolated into an MDA (Assay Kit Sigma-Aldrich, Darmstadt, Germany) standard curve (0–3 mg/g wet tissue).

### 2.3. Histology

For histological analysis, EXP55, CTRL55, EXP85, and CTRL85 animals (2 per group) were anesthetized intraperitoneally with xylazine (Rompum 10 mg/kg; BAYER, Leverkusen, Germany) and subsequently with pentobarbital (120–210 mg/kg; MSD, Rahway, NJ, USA). The animals were subsequently perfused with saline (0.9% NaCl; PiSA, México city, México) and paraformaldehyde (4%; Thermo Fisher Scientific, Leicestershire, UK) diluted in 0.1 M phosphate buffer [PB], pH 7.4. The brains were dissected and postfixed in 4% paraformaldehyde for 24 h. The brains were then embedded in paraffin, and coronal sections (5 μm thick) were cut every 100 μm via a microtome (Leica RM2125RT, Wetzlar, Germany). The obtained sections were placed in a gelatin flotation bath, mounted on slides, and stained with cresyl violet (Sigma–Aldrich, Darmstadt, Germany) to visualize the cell bodies and histological evidence of damage through all areas of the hippocampus, amygdala, and PCX via an Olympus BX51 brightfield microscope (Evident Scientific, Tokyo, Japan).

### 2.4. Statistical Analysis

All the data are expressed as the means ± standard deviations (SDs) for the animals in each group. To determine differences between GSH and GSSG levels, the GSH/GSSG ratio, and MDA levels, a Student’s *t*-test for independent samples was used after checking for homoscedasticity and normality via a Q–Q plot analysis and a Shapiro–Wilk test. A *p*-value < 0.05 was considered statistically significant.

## 3. Results

The GSH activity levels measured for CTRL55 and EXP55 in the PCX group and for CTRL55 and EXP55 in the LIMB group were 8.082 ± 0.227, 6.449 ± 1.051, 7.290 ± 0.619, and 6.852 ± 0.820 nmol of GSH/mg of protein, respectively. The GSH activity in PCX was significantly greater (by 1.25-fold) in the CTRL55 group than in the EXP55 group (*p* = 0.015). No significant changes in the LIMB were observed in the CTRL55 and EXP55 groups (Figure 2A). The GSH activity data for the CTRL85 and EXP85 groups in PCX and the CTRL85 and EXP85 groups in LIMB were 3.850 ± 0.100, 2.718 ± 0.241, 6.765 ± 0.183, and 4.048 ± 0.421 nmol of GSH/mg of protein, respectively. In PCX, GSH activity in the CTRL85 group was significantly greater (by 1.4-fold) than that in the EXP85 group (*p* = 0.0001); likewise, in LIMB, GSH activity in the CTRL85 group was significantly greater (by 1.7-fold) than that in the EXP85 group (*p* = 0.0001) (Figure 2B).

The GSSG activities determined for the groups CTRL55 and EXP55 in PCX and CTRL55 and EXP55 in LIMB were 0.698 ± 0.012, 0.776 ± 0.026, 0.202 ± 0.028, and 0.473 ± 0.111 nmol of GSSG/mg of protein, respectively. GSSG activity in the PCX was significantly greater (by 1.11-fold) in the EXP55 group than in the CTRL55 group (*p* = 0.0003). In addition, in LIMB, GSSG activity in the EXP55 group was significantly greater (2.3-fold) than that in the CTRL55 group (*p* = 0.0008) (Figure 2C). The GSSG activity levels for the CTRL85 and EXP85 groups in PCX and the CTRL85 and EXP85 groups in LIMB were 0.190 ± 0.015, 0.434 ± 0.114, 0.177 ± 0.063, and 0.334 ± 0.031 nmol of GSSG/mg of protein, respectively (Figure 2D).

In PCX, GSSG activity in the EXP85 group was significantly greater (2.28-fold) than that in the CTRL85 group (*p* = 0.0054); likewise, in LIMB, GSSG activity in the EXP85 group was significantly greater (1.89-fold) than that in the CTRL85 group (*p* = 0.0195) (Figure 2D).

The biochemical ratios were calculated from the measured parameters. The GSH/GSSG ratios in the CTRL55, EXP55 in PCX, and CTRL55 and EXP55 in the LIMB groups were 11.578 ± 0.535, 9.073 ± 3.178, 49.852 ± 3.855, and 15.035 ± 3.469, respectively. In the PCX group, the GSH/GSSG ratios in the CTRL55 and EXP55 groups were very similar, with no significant differences; however, in the LIMB group, the GSH/GSSG ratio was significantly greater in the CTRL55 group than in the EXP55 group (3.31-fold) (*p* = 0.0001) (Figure 2E). Moreover, the GSH/GSSG ratios in the CTRL85, EXP85 in PCX, and CTRL85 and EXP85 in the LIMB groups were 24.16 ± 8.951, 14.37 ± 2.007, 21.08 ± 5.048, and 9.702 ± 2.282, respectively. For PCX, the GSH/GSSG ratio in the CTRL85 group was greater than that in the EXP85 group (1.68-fold); however, this difference was not statistically significant. In the LIMB group, the GSH/GSSG ratio was significantly greater in the CTRL85 group than in the EXP85 group (2.17-fold) (*p* = 0.0063) (Figure 2F).

Furthermore, concerning the lipid peroxidation levels measured for CTRL55 and EXP55 in the PCX group and CTRL55 and EXP55 in the LIMB group, the values were 5.327 ± 0.458, 13.762 ± 3.263, 4.810 ± 0.512, and 4.917 ± 0.759 nmol of MDA/mg of protein, respectively. Lipid peroxidation in PCX was significantly greater (2.58-fold greater) in the EXP55 group than in the CTRL55 group (*p* = 0.0005). No significant changes in the LIMB were observed in the CTRL55 and EXP55 groups (Figure 3A). The lipid peroxidation levels for the CTRL85 and EXP85 groups in PCX and the CTRL85 and EXP85 groups in LIMB were 1.674 ± 0.391, 2.330 ± 0.652, 1.507 ± 0.376, and 2.365 ± 0.806 nmol of MDA/mg protein, respectively. In PCX, lipid peroxidation in the EXP85 group was significantly greater (1.39-fold greater) than that in the CTRL85 group (*p* = 0.037); in addition, in LIMB, lipid peroxidation in the EXP85 group was significantly greater (1.57-fold greater) than that in the CTRL85 group (*p* = 0.0017) (Figure 3B).

The histology study revealed the following results: Figure 4 shows micrographs of the PCX of the brains of the CTRL55 and CTRL85 groups (4A and 4C) and the EXP55 and EXP85 groups (4B and 4D). Panels 4A and 4C show cells with a normal cytoarchitecture, a conserved shape, and evident nuclei and nucleoli; moreover, compared with the EXP groups, the CTRL groups presented fewer shrinking cells. Panels 4B and 4D show the presence of angular and pyknotic cells, as well as shrinking cells. The number of damaged cells is slightly greater in the EXP85 group than in the EXP55 group. The histological damage observed in the EXP groups occurred at 55 days of age and was maintained until 85 days of age. Its manifestation begins with the appearance of a moderate number of angular and pyknotic cells, in which the nucleoli are not evident. In the EXP85 group, evident cellular degeneration was observed, characterized by a lack of visible nuclei, a reduction in cell size, and increased acidophilia with dead cell debris.

Figure 5 shows micrographs of the hippocampus of the brains of the CTRL55 and CTRL85 groups (5A and 5C), which revealed cells with normal cytoarchitecture and shapes with obvious nuclei and nucleoli. In the EXP55 group, tissue disorganization and the presence of shrinking and pyknotic cells with a notable absence of nuclei and nucleoli were observed (Figure 5B). In the EXP85 group, there was marked tissue disorganization, with many pyknotic, fusiform, and shrinking cells with pronounced acidophilia (Figure 5D).

Figure 6 shows micrographs of the amygdala of animals in groups CTRL55 and CTRL85 (6A and 6C) and EXP55 and EXP85 (6B and 6D). Panels 6A and 6C show cells with normal cytoarchitecture, conserved shapes, and evident nuclei and nucleoli; the cell distribution is homogeneous, although a few cells in the process of degeneration appear. Panel 6B shows the amygdala micrograph of the EXP55 animals; disorganization of the tissue is observed with the presence of shrinking and pyknotic cells with a notable absence of nuclei and nucleoli. Additionally, many preserved cells remain. Panel 6D shows sparse conserved cells and abundant pyknotic and shrunken cells with increased acidophilic staining. Cellular disorganization was observed, and the number of damaged cells was greater than that in the EXP55 group.

## 4. Discussion

In this study, the following important findings were observed: (1) The effects of CLO on GSH levels in the PCX and LIMB were more significant in rats at 85 days of age, as significant differences were observed in both brain structures compared with those in the CTRL group. At 55 days of age, only significant changes in GSH levels were observed in the PCX. At both time points, GSH levels were lower in the PCX of the EXP groups than in those of the CTRL groups. However, compared with those in the CTRL groups, the GSSG levels differed significantly in both brain structures at both ages. In contrast to GSH, GSSG levels increased in both brain structures in the EXP groups at both ages. A significant decrease in the GSH/GSSG ratio was found in the LIMB of the EXP groups compared with that in the CTRL groups at both ages. (2) About the MDA levels, an accentuated effect was observed in both brain structures after 85 days of age, with a significant increase observed in the EXP group compared with the CTRL group. After 55 days of age, however, only a significant increase in PCX was observed compared with that in the CTRL group. (3) Histological analysis revealed that, in the PCX and LIMB (hippocampus and amygdala) of the CTRL55 and CTRL85 groups, cells with normal cytoarchitecture, a conserved shape, and evident nuclei and nucleoli were observed. In addition, the cell distribution was homogeneous, with the presence of sparse cells during the degeneration process. Conversely, in the PCX, the hippocampus and amygdala of the EXP85 group marked tissue disorganization, and the presence of shrinking and pyknotic cells with an evident absence of nuclei and nucleoli was observed; the same patterns were detected in the EXP55 group but to a lesser extent.

These findings are consistent with previous studies demonstrating that CLO administration is associated with increased OS. A recent study revealed that chronic daily administration of CLO (2.25 mg/kg) for eight weeks increased OS and inflammation levels in the testes, liver, and heart of adult rats [25]. This study revealed that GSH levels and SOD activity were significantly reduced, whereas MDA and tumor necrosis factor levels were significantly increased in these three organs compared with those in the CTRL group [25]. Another study in rats revealed that daily administration of a high dosage of CLO (75 mg/kg) for six days significantly increased the hydrogen peroxide (H_2_O_2_) concentration in the liver and significantly decreased SOD activity in the PCX compared with that in the CTRL group [26]. In the present study, we observed the same effects on GSH and MDA levels in two brain structures (significantly in the PCX at 55 and 85 days) of rat pups that were chronically administered CLO at high doses, demonstrating that the drug impacts the cellular redox state over time by inducing an oxidative state. In terms of its pharmacokinetics, CLO is absorbed from the gastrointestinal tract and is mainly demethylated at the hepatic level by several cytochrome P450 (CYP) enzymes, including 1A2, 2C19, 3A4, and 21A2, to its active metabolite desmethylclomipramine [27,28,29,30]. CLO is distributed throughout the cerebrospinal fluid and crosses the blood–brain barrier, generating oxidative damage to lipid membrane structures and leading to increased MDA levels and decreased GSH levels [31,32]. Conversely, studies have demonstrated differential distribution of the drug in the rat brain, with the highest concentrations found in the PCX, amygdala, thalamus, and striatum compared with the hypothalamus, olfactory bulb, septum, and cerebellum after 4 h of intravenous administration (10 mg/kg) [33]. Similar results were observed in rats that were administered with CLO (15 mg/kg) for 14 days, with the highest concentrations found in the PCX and successively lower concentrations in the hypothalamus, striatum, cerebellum, hippocampus, and brainstem [34]. The higher concentration of CLO in the PCX and the subsequent greater pharmacological effect could explain the significant changes in GSH and MDA levels in the PCX compared with those in the LIMB, particularly at 55 days of age. In relation to GSSG levels, we observed an increase in the PCX of CTRL55 in comparison with that of CTRL85, which could be explained by GSH metabolism being dependent on factors such as aging or disease conditions [35]. In postnatal rat brain development, changes in GSH levels in the cortex and cerebellum have been observed, as intense synaptogenesis during the first postnatal stages is significantly increased at postnatal day 7 [36]. Another reason is the homeostasis glutathione disturbance induced by manipulation in rats at an early age, when the nervous system is particularly susceptible and dependent on the GSH content in the development stage [37].

The chemical structure of the drug contains two benzene rings, which are biotransformed by CYP2E1 into various phenolic metabolites, such as hydroquinone and benzenetriol, that induce the generation of ROS that can interact with cellular DNA, inducing the presence of 8-hydroxy-2′-deoxyguanosine and other hydrophobic aromatic-DNA adducts [38,39,40]. Moreover, these products may undergo a process after penetrating the cell, which leads to an increase in toxicity, the formation of electrophilic metabolites, and lipid peroxidation [40]. The presence of reactive electrophilic metabolites, such as lipid-derived electrophiles (LDEs), is relevant in the central nervous system because they have been implicated in neurodegenerative diseases [41]. These findings suggest that the presence of LDEs derived from oxidative stress caused by the oxidant CLO affects the abundance of polyunsaturated fatty acids in the brain during the neonatal stages in rats, resulting in structural changes in different brain structures. On the other hand, the presence of LDEs can regulate Keap1/Nrf2/antioxidant response signaling, which could affect the synthesis of antioxidants such as the GSH system in our rat model [42].

Another study in rats revealed that, compared with the CTRL, CLO increased lipid oxidation levels and significantly decreased GSH levels in the heart [15]. Furthermore, through an in vitro deoxyribose assay, this study demonstrated that CLO significantly increases hydroxyl radical (HO•) levels [15]. HO• is the most reactive oxidizing species, suggesting that the ability of CLO to generate this radical could explain its ability to oxidize and damage the neuronal membrane. This extreme reactivity is due to its small size and lack of charge, allowing rapid diffusion across cellular structures and causing extensive cellular damage through processes such as lipid peroxidation and DNA strand breaks [43]. CLO degrades DNA in the presence of peroxidases and H_2_O_2_ [44,45]. On the other hand, HO• has been shown to induce neuronal and glial death with features characteristic of both necrosis and apoptosis [46]. These observations are consistent with results from a study in which antidepressants such as fluoxetine, sertraline, and CLO were found to be cytotoxic to rat blood–brain barrier cells. This occurred by reducing the mitochondrial membrane potential and the activities of mitochondrial complexes I and III, as well as by increasing ROS levels at high drug concentrations (1–100 μM) [47]. CLO is recognized as a compound that obstructs mitochondrial function [48,49]. In the liver, gluconeogenesis and ureagenesis, processes that depend on the mitochondrial ATP-inhibiting ATP synthase complex and electron flow, are inhibited [16]. In recent research, the bioactive metabolite CLO desmethylclomipramine was shown to cause transforming growth factor (TGF)-β1-mediated mesenchymal-type A549 cells to undergo mitochondrial death via myeloid cell leukemia-1 (Mcl-1) suppression and activation of truncated Bid (tBid). The drug induces mitochondrial instability and caspase activation through inactivation of the Akt/GSK-β/Mcl-1 axis [50]. Moreover, CLO inhibits growth and inhibits genes related to OS defense (thioredoxin peroxidase, glutaredoxin, nitric oxide oxidoreductase, cytochrome c peroxidase and b5 reductase, GPx, SOD1, SOD2, and CAT) in glioblastoma patient biopsy-derived cell cultures [51]. This evidence suggests that CLO treatment results in OS and that mitochondrial ROS compromises the first line of antioxidant defenses in the brain structures of our rat model.

Furthermore, a proteomic analysis of the PCX of 8–21-day-old neonatal rats treated with a high dose of CLO (20 mg/kg twice daily) revealed that, compared with the CTRL, the drug induced changes in the expression of proteins in the PCX that are key to energy metabolism and mitochondrial function. CLO increases the expression of galactokinase and triose phosphate isomerase while reducing the expression of glyceraldehyde-3-phosphate dehydrogenase and phosphoglycerate kinase, which are key enzymes in glycolysis. Additionally, the drug was found to reduce the levels of aconitase 2 and pyruvate dehydrogenase, which play pivotal roles in the tricarboxylic acid cycle. The NADH dehydrogenase levels decreased, whereas the ATP synthase levels increased with treatment. These proteins are linked to energy production and restoration through their role in oxidative phosphorylation. Furthermore, creatine kinase B levels increased. This protein is also important for mobilizing energy supplies in the brain. CLO modulates the expression of proteins related to cytoskeletal plasticity, the stress response, transcription and translation, and immunomodulation. The authors concluded that chronic CLO administration to neonates induces molecular-level changes and that treatment of a flexible neonatal brain initiates complex epigenetic and other developmental changes that modify the proteome [52]. These findings related to the effects of CLO at the biochemical and molecular levels suggest that administering CLO at high doses chronically can induce changes in energy levels and cellular redox rates in the central nervous system when it is administered from the neonatal stage. These changes explain the increase in lipid oxidation levels, decrease in GSH levels, and increase in GSSG levels observed in PCX and LIMB, which were more significant in PCX.

The administration of high doses during early development in young rats resulted in persistent and multifaceted alterations within the PCX, hippocampus, and amygdala. These changes may influence structural, functional, and neurochemical signaling in regions integral to emotional and cognitive regulation [53]. Neonatal CLO exposure induced sustained modifications in the proteome of PCX, notably affecting proteins associated with inflammation, metabolism, transcription, and cytoskeletal organization [52]. Specifically, alterations in the expression of the macrophage migration inhibitory factor protein, which is linked to inflammatory processes and depressive symptoms, have been observed [52]. These findings indicate that neuronal and glial development may be adversely affected, potentially contributing to lasting cell damage [52]. Collectively, these data demonstrate that early exposure to CLO leads to enduring alterations in brain architecture and function. Neuroinflammation, hormonal receptor reorganization, and protein expression changes associated with such exposure appear to increase vulnerability to cellular damage [54,55,56].

Finally, CLO, a nonselective monoamine reuptake inhibitor, interferes with the function of SERT or NETs. CLO, in its original form, shows a significantly stronger affinity for SERT than for NET. The sustained increase in monoamines necessitates subsequent metabolism to maintain homeostasis, a process that relies heavily on the enzyme monoamine oxidase (MAO). Oxidative deamination, catalyzed by MAO, has been shown to generate H_2_O_2_ as a byproduct. This ROS has been demonstrated to be capable of overwhelming the body’s antioxidant defenses and triggering OS in excess [57].

The prolonged and sustained increase in serotonin and norepinephrine levels in the synaptic space is the desired effect of clomipramine. However, the homeostasis of these neurotransmitters does not depend solely on reuptake. The body has enzymatic degradation mechanisms that act as a second line of control to regulate monoamine concentrations. Therefore, a sustained increase in the substrate for this metabolic pathway, such as that produced by chronic inhibition of transporters, has the potential to overload enzymatic capacity and alter cellular biochemistry [20,21]. The reviewed findings establish a plausible mechanistic link, supported by experimental evidence in animals, between the pharmacological effect of CLO and the induction of OS. The sequence of events, inhibition of serotonin and norepinephrine reuptake, increased concentration in the synapse, increased metabolism via the MAO pathway, and consequent production of H_2_O_2_, provides a coherent explanation for the increase in markers of oxidative damage observed in the rat models (Figure 7).

## 5. Conclusions

This study revealed that subchronic administration of CLO during the neonatal period causes lasting biochemical and morphological changes in the PCX and LIMB of rats into adulthood. These alterations reflect a redox imbalance in the brain characterized by increased OS and persistent cellular damage, manifested as increased biomarkers such as GSSG and MDA, in addition to histological alterations, including the accumulation of pyknotic cells and tissue degradation. These findings go beyond the mere observation of oxidative damage, indicating that early exposure to this drug can trigger neurotoxic and neuroinflammatory processes that may compromise the structural and functional integrity of areas linked to emotional and cognitive processing.

Notably, this study has several limitations, particularly the inclusion of a larger number of markers of oxidative damage, specifically mitochondrial dysfunction, and the evaluation of the drug’s effect on animal behavior. In the future, it may be possible to conduct studies in rats of different ages, equivalent to pediatric age in humans, and analyze the effects of different doses of CLO.

## Figures and Tables

**Figure 1 brainsci-15-01068-f001:**
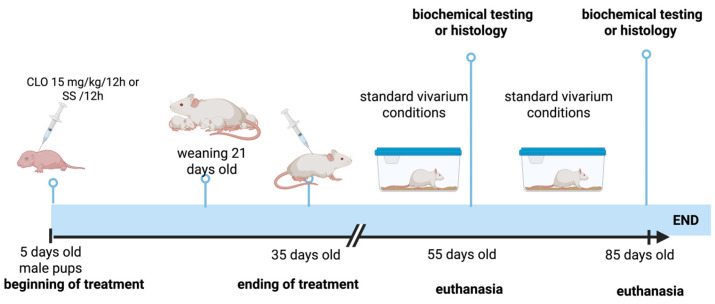
Chronological representation of the experimental design.

**Figure 2 brainsci-15-01068-f002:**
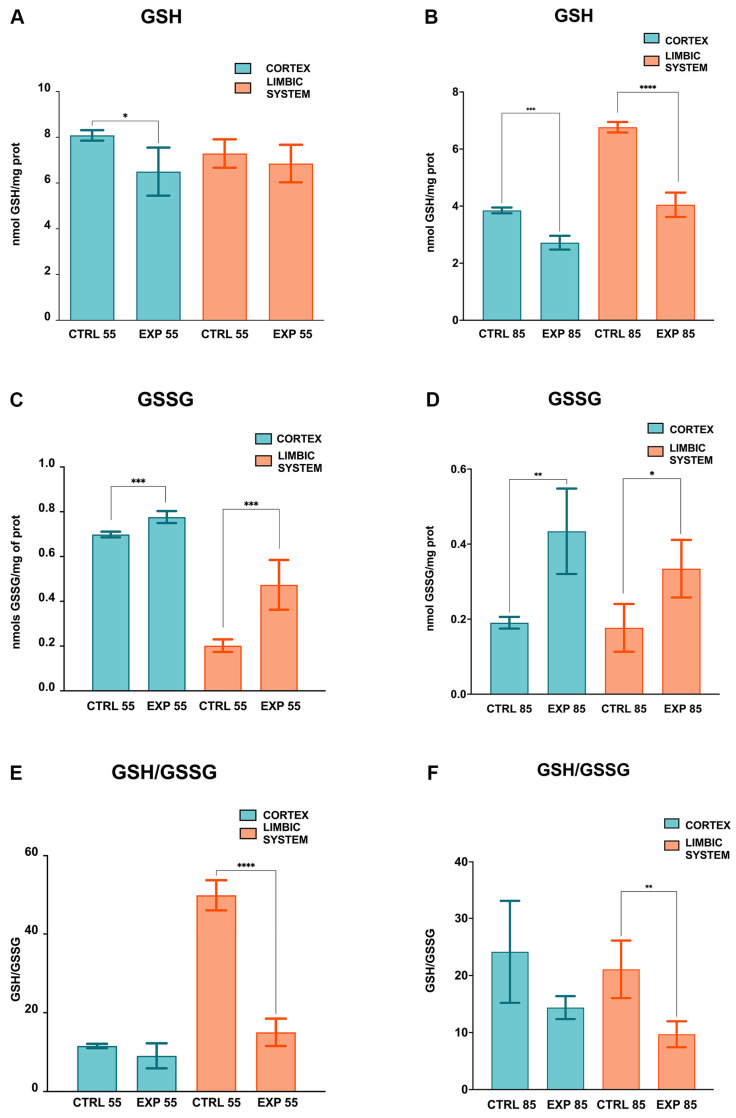
Glutathione levels in the cortex and limbic system (hippocampus and amygdala) of rats treated with clomipramine. GSH levels in the prefrontal cortex (PCX) and limbic system (LIMB) of control (CTRL) and experimental (EXP) rats at 55 (**A**) and 85 (**B**) days of age. GSSG levels in the PCX and LIMB of CTRL and EXP rats at 55 (**C**) and 85 (**D**) days of age. GSH/GSSG metabolic rate in the PCX and LIMB of CTRL and EXP rats at 55 (**E**) and 85 (**F**) days of age. All the measurements were performed in triplicate. The values are presented as the means ± SDs (n = 8); differences were analyzed via Student’s t test for independent samples. * *p* < 0.05, ** *p* < 0.01, *** *p* < 0.001, and **** *p* < 0.0001.

**Figure 3 brainsci-15-01068-f003:**
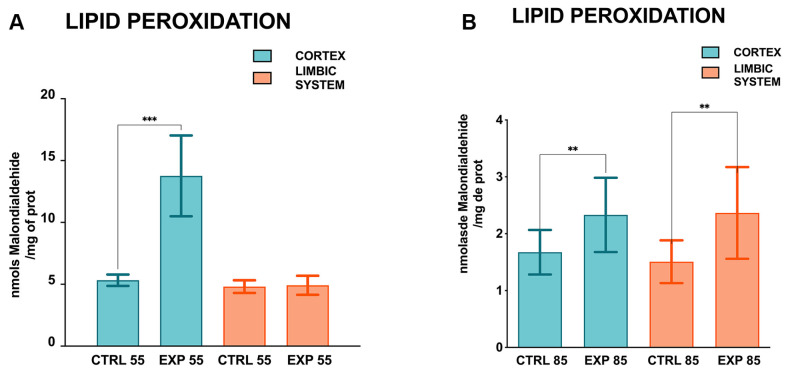
Lipid peroxidation levels in the cortex and limbic system (hippocampus and amygdala) of clomipramine-treated rats. Malondialdehyde levels in the prefrontal cortex and limbic system of control (CTRL) and experimental (EXP) rats at 55 (**A**) and 85 (**B**) days of age. All measurements were made in triplicate following the TBARS technique. The values are presented as the means ± SDs (n = 8); differences were analyzed via Student’s t-test for independent samples. ** *p* < 0.01, and *** *p* < 0.001.

**Figure 4 brainsci-15-01068-f004:**
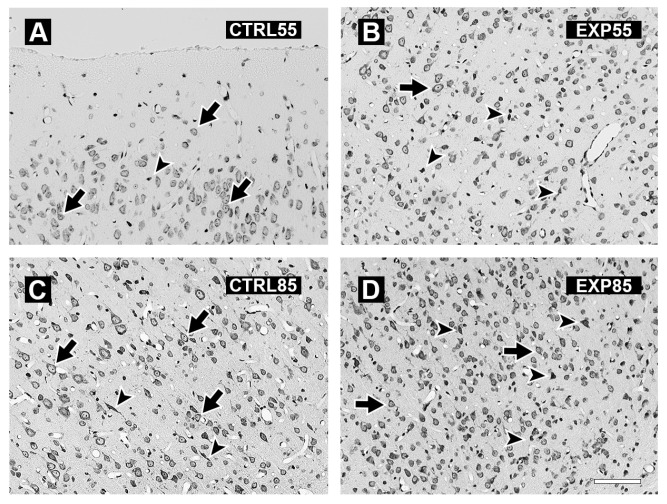
Photomicrographs of representative coronal sections of the prefrontal cortex of control (CTRL) and clomipramine-treated (EXP) rats. Panels (**A**,**C**) show sections from CTRL55 and CTRL85 rats, respectively. Panels (**B**,**D**) display sections from EXP55 and EXP85 rats, respectively. In panels (**A**,**C**), intact cells, as indicated by arrows, are visible; some shrinking cells (head arrows) can also be observed. Conversely, panels (**B**,**D**) present a greater number of damaged cells (arrowheads) and a smaller number of normal cells (arrows). Scale bar: 40 µm.

**Figure 5 brainsci-15-01068-f005:**
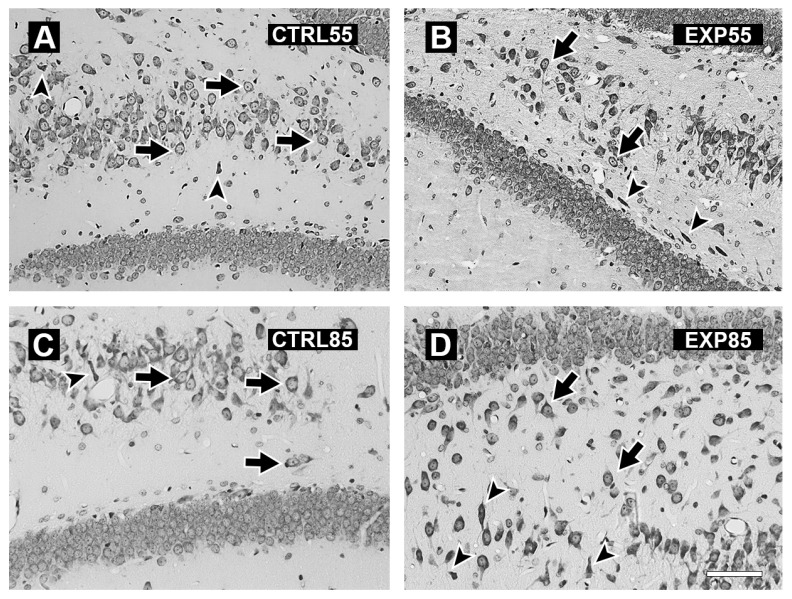
Photomicrographs showing representative coronal sections of the hippocampus of control (CTRL) and clomipramine-treated (EXP) rats. Panels (**A**,**C**) display hippocampal sections from CTRL55 and CTRL85 rats, respectively. Panels (**B**,**D**) present hippocampal sections from EXP55 and EXP85 rats, respectively. In panels (**A**,**C**), the cells exhibit normal cytoarchitecture (arrows) and sparsely shrinking cells (head arrows). In contrast, panels (**B**,**D**) reveal tissue disorganization and a great number of damaged cells (head arrows), as well as normal cells (arrows). Scale bar: 40 µm.

**Figure 6 brainsci-15-01068-f006:**
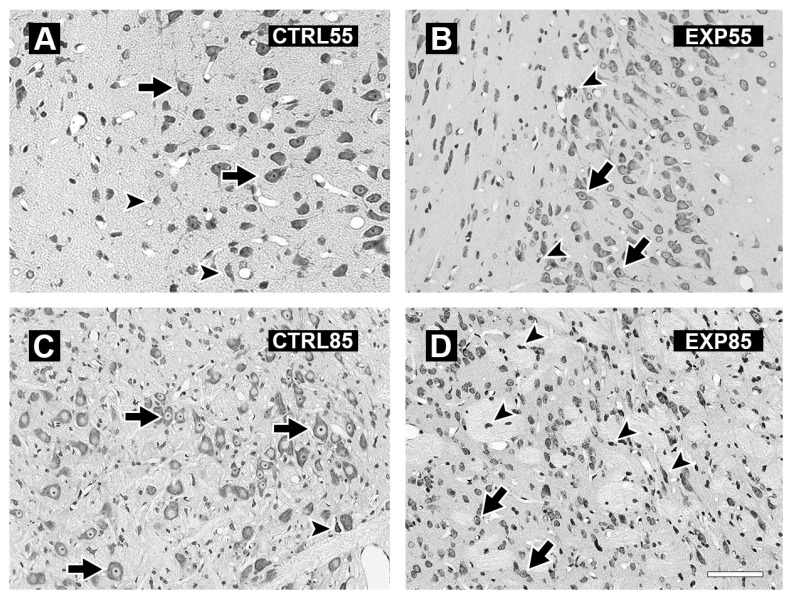
Photomicrographs showing representative coronal sections of the amygdala of control (CTRL) and clomipramine-treated (EXP) rats. Panels (**A**,**C**) present amygdala sections from CTRL55 and CTRL85 rats, respectively. Panels (**B**,**D**) display comparable sections from EXP55 and EXP85 rats, respectively. Panels (**A**,**C**) reveal cells with normal cytoarchitecture (indicated by arrows) and sparsely shrinking cells (head arrows). In contrast, Panel (**B**) shows tissue disorganization with shrinking cells (head arrows); there are still preserved cells (arrows). Panel (**D**) shows many damaged cells (head arrows), with few normal cells (arrows). Scale bar: 40 µm.

**Figure 7 brainsci-15-01068-f007:**
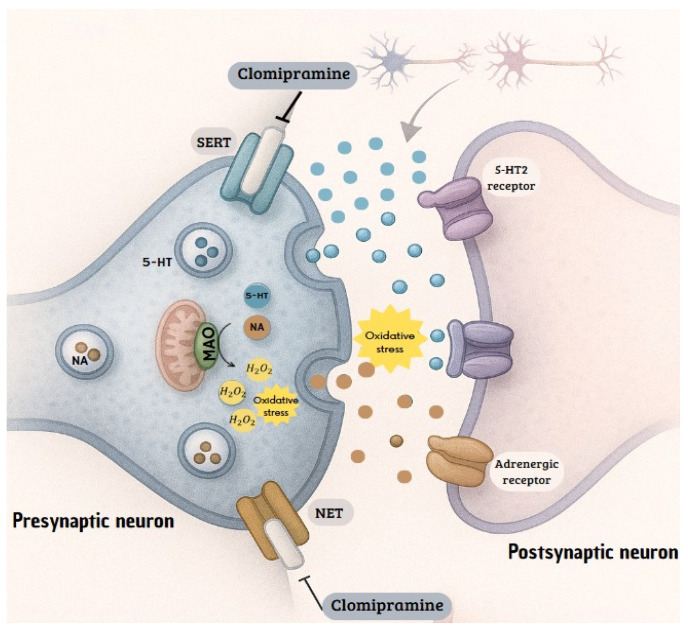
Mechanism of action of CLO in inhibiting the reuptake of serotonin and norepinephrine. The mechanism of action of CLO involves the suppression of the reuptake of monoamines, primarily serotonin and norepinephrine. This is achieved by inhibiting the transporters SERT (serotonin transporter) and NET (norepinephrine transporter). This causes the accumulation of these amines in the synaptic cleft and presynaptic terminal. One possible explanation for how CLO induces oxidative stress is that monoamine oxidase (MAO) degrades the accumulated bioamines in the presynaptic terminal; during this process, hydrogen peroxide, a precursor to reactive oxygen species (ROS), is generated.

## Data Availability

The data presented in this study are available upon request from the corresponding author.

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
