# Peer review of "Clomipramine Induced Oxidative Stress and Morphological Alterations in the Prefrontal Cortex and Limbic System of Neonatal Rats"

_brainsci, 2025, doi:10.3390/brainsci15101068_

Round 1

Reviewer 1 Report

Comments and Suggestions for Authors

The manuscript “Clomipramine induced oxidative stress and morphological alterations in the prefrontal cortex and limbic system of neonatal rats” written by Norma Angélica Labra-Ruíz and coauthors report on the study in which the subchronic administration of clomipramine into neonatal male rats results in the persistence of oxidative stress in prefrontal cortex and limbic system on later stages of maturation. Authors have shown that administration of clomipramine at high doses into the rats from 5 to 35 days increases the levels of oxidized glutathione and malondialdehyde in both brain regions at 55 and 85 days. Undoubtedly, authors touch upon a topical issue concerning into the effects of widely used antidepressants on various brain structures that is one of the important directions in neuroscience due to the complexity of understanding the mechanisms of brain responses and how they can be changed under the used drugs. The manuscript is written in satisfactory English language and generally agrees the requirement of Brain Sciences. However, this manuscript requires serious revision, in particular some additional tests are needed to confirm the effect of clomipramine on maintaining oxidative stress over a long period of time. Some issues that should to be addressed are below.

Major comments

  1. Why did the authors choose days 55 and 85 for oxidative stress analysis?
  2. According to this study, the authors established the levels and ratio of reduced and oxidized forms of glutathione and the level of malondialdehyde. As known, one of the main factors of oxidative stress is reactive oxygen species, which level was not determined in this study but in the Discussion section authors speculated that clomipramine might increase hydroxyl radical level in vitro. In my opinion, authors should presented the ROS formation level in the brain regions studied in the manuscript.

In addition, the level of activity and/or expression levels of superoxide dismutase and catalase are one more oxidative stress markers. Despite the authors mention that the activity of these enzymes has already been determined in previous studies, they should have assessed the activity of these enzymes in the brain regions.

  1. How can the authors explain that the level of GSSG in Ctrl55 group is higher than in Ctrl85 in cortex region?
  2. Despite the small number of experiments presented in the manuscript, the authors did a great job. Did the authors observe the rats' behavior throughout the experiment? If so, these results could improve the manuscript.

Minor comments

  1. In the Introduction section, the paragraph describing the mechanism of clomipramine action should be added for better understanding.
  2. All materials and equipment should have the manufacturers’ name and country in the Materials and Method section.
  3. In the Results section, all numbers should be deleted from the main text and formatted into a table in Supplementary file due to they make difficulties in understanding the results obtained.
  4. To minimize empty lines in the main text Figure 1 should be moved after paragraph 1 of the Results section.
  5. The frames on Figure 1 have different thicknesses. It is necessary to bring them to uniformity. Moreover, descriptions of cell states should be deleted from the captions of Figure 3-5 because they repeated in the main text.
  6. Line 212: a word “respectively” should be inserted at the end of the sentence.
  7. Line 273: apparently, “CTRL55” should be changed to “Exp55”.
  8. Lines 276-278: references on the studies mentioned should be inserted.
  9. Lines 305-306: references on the study mentioned should be inserted.

In conclusion, I recommend this manuscript to be published in Brain Sciences after major revision.

Author Response

Reviewer 1

The manuscript “Clomipramine-induced oxidative stress and morphological alterations in the prefrontal cortex and limbic system of neonatal rats,” written by Norma Angélica Labra-Ruíz and coauthors, reports on the study in which the subchronic administration of clomipramine into neonatal male rats results in the persistence of oxidative stress in the prefrontal cortex and limbic system at later stages of maturation. Authors have shown that administration of clomipramine at high doses to rats from 5 to 35 days increases the levels of oxidized glutathione and malondialdehyde in both brain regions at 55 and 85 days. Undoubtedly, authors touch upon a topical issue concerning the effects of widely used antidepressants on various brain structures that is one of the important directions in neuroscience due to the complexity of understanding the mechanisms of brain responses and how they can be changed under the used drugs. The manuscript is written in satisfactory English and generally agrees with the requirements of Brain Sciences. However, this manuscript requires serious revision; in particular, some additional tests are needed to confirm the effect of clomipramine on maintaining oxidative stress over a long period of time. Some issues that should be addressed are below.

First, we would like to thank the reviewer for his/her comments. Below, we address each of the points with the hope that the answers are satisfactory to you.

  1. Why did the authors choose days 55 and 85 for oxidative stress analysis?

A= The purpose of analyzing the effect of clomipramine at both ages would allow us to identify whether the damage caused by drug administration in newborns was sustained over time or whether compensatory mechanisms were evident during puberty and young adulthood. Importantly, in rats, day 55 of age marks the beginning of puberty, a stage that coincides with intense neuronal and hormonal activity where the maturation of neuronal circuits responsible for decision-making, risk responses, and emotional management favors the development of skills, adaptations to the environment, and new learning. Day 85 of age is equivalent to the transitional stage between adolescence and adulthood (Sengupta 2013). Although neuronal plasticity decreases compared with that in the previous stage, especially in the limbic system, it marks the final maturation period in the frontal cortex and allows the brain to modify itself at the structural and functional levels, allowing functional consolidation and recovery from injuries (Statsenko et al., 2025). We added part of this answer to the materials and methods section (lines 86-88).

  1. According to this study, the authors established the levels and ratio of reduced and oxidized forms of glutathione and the level of malondialdehyde. As is known, one of the main factors of oxidative stress is reactive oxygen species, whose level was not determined in this study, but in the Discussion section, the authors speculated that clomipramine might increase hydroxyl radical level in vitro. In my opinion, authors should present the ROS formation level in the brain regions studied in the manuscript. In addition, the level of activity and/or expression levels of superoxide dismutase and catalase are one more oxidative stress marker. Despite the authors’ mention that the activity of these enzymes has already been determined in previous studies, they should have assessed the activity of these enzymes in the brain regions.

A= It is acknowledged that there are additional oxidative stress markers that may be suitable for this study. We appreciate your suggestion; however, as this is a brief communication, we believe that the findings are valuable enough to be shared. With these first results, in future research, we could expand the study of oxidative stress markers and measure ROS, SOD, catalase, and glutathione peroxidase, as well as nitrosative stress.

  1. How can the authors explain that the level of GSSG in the Ctrl55 group is higher than in Ctrl85 in the cortex region?

A= We explain this finding as GSH metabolism being dependent on factors such as aging or disease conditions. During postnatal rat brain development, changes in GSH levels in the cortex and cerebellum have been observed, as intense synaptogenesis during the first postnatal stages is significantly increased at postnatal day 7. Another reason is the homeostasis glutathione disturbance induced by manipulation in rats at an early age, when the nervous system is particularly susceptible and dependent on the GSH content in the development stage. We added this explanation to the discussion section (Lines 366--374).

  1. Despite the small number of experiments presented in the manuscript, the authors did a great job. Did the authors observe the rats' behavior throughout the experiment? If so, these results could improve the manuscript.

A= As a brief communication, the objective of the present study was to determine the effects of subchronic administration of clomipramine on oxidative stress and whether there was any type of morphological damage. However, in the future, we could perform behavioral tests, such as the open field test and the forced swim test, and use the rotarod test to evaluate whether motor problems can affect the results of these tests.

Minor comments

  1. In the Introduction section, the paragraph describing the mechanism of clomipramine action should be added for better understanding.

A= In accordance with your request, we added a paragraph describing the mechanism of action of clomipramine (Lines 40--47).

  1. All materials and equipment should have the manufacturer’s name and country in the Materials and Methods section.

A= In accordance with your request, we added the name and country of the materials and equipment tothe Materials and Methods section. 

  1. In the Results section, all numbers should be deleted from the main text and formatted into a table in the supplementary file because they make it difficult to understand the results obtained.

A= Thank you for your comment. Although we understand your point of view, it is our custom and preference to describe the data in detail to keep the results complete and accessible to other researchers and ourselves. We have described the results this way in at least three other previously published papers (Ignacio-Mejía et al., 2023; Contreras-García et al., 2024; Ignacio-Mejía et al., 2024). However, if you insist, we can rewrite the results and create a supplementary table.

  1. To minimize empty lines in the main text, Figure 1 should be moved after paragraph 1 of the Results section.

A= In accordance with your request, we have restructured the order of the paper to avoid empty lines.

  1. The frames in Figure 1 have different thicknesses. It is necessary to bring them to uniformity. Moreover, descriptions of cell states should be deleted from the captions of Figures 3-5 because they are repeated in the main text.

A= According to your request, we made the relevant modifications.

  1. Line 212: The word “respectively” should be inserted at the end of the sentence.

A=The requested change was made (lines 266, 284, and 313).

  1. Line 273: apparently, “CTRL55” should be changed to “Exp55.”

A= Thank you for your comment; indeed, CTRL55 was incorrect. The correction for EXP55 was made (line 337).

  1. Lines 276-278: References to the studies mentioned should be inserted.

A= In accordance with your request, we added the references to the manuscript (line 342).

  1. Lines 305-306: References to the study mentioned should be inserted.

A= In accordance with your request, we added these references to the manuscript (390).

References

  1. Sengupta, P. (2013). The Laboratory Rat: Relating Its Age With Humans. International Journal of Preventive Medicine, 4(6), 624–630.
  2. Statsenko, Y., Kuznetsov, N. V., & Ljubisaljevich, M. (2025). Hallmarks of Brain Plasticity. Biomedicines, 13(2), 460. https://doi.org/10.3390/biomedicines13020460
  3. Ignacio-Mejía, I., Contreras-García, I. J., Mendoza-Torreblanca, J. G., Medina-Campos, O. N., Pedraza-Chaverri, J., García-Cruz, M. E., Romo-Mancillas, A., Gómez-Manzo, S., Bandala, C., Sánchez-Mendoza, M. E., Pichardo-Macías, L. A., & Cárdenas-Rodríguez, N. (2023). Evaluation of the Antioxidant Activity of Levetiracetam in a Temporal Lobe Epilepsy Model. Biomedicines, 11(3), 848. https://doi.org/10.3390/biomedicines11030848.
  4. Contreras-García, I. J., Bandala, C., Ignacio-Mejía, I., Pichardo-Macías, L. A., Mendoza-Torreblanca, J. G., García-Cruz, M. E., Gómez-Manzo, S., & Cárdenas-Rodríguez, N. (2024). Effect of levetiracetam on DNA oxidation and glutathione content in a temporal lobe epilepsy model. Acta neurobiologiae experimentalis, 84(1), 51–58. https://doi.org/10.55782/ane-2024-2537.
  5. Ignacio-Mejía, I., Contreras-García, I. J., Pichardo-Macías, L. A., García-Cruz, M. E., Ramírez Mendiola, B. A., Bandala, C., Medina-Campos, O. N., Pedraza-Chaverri, J., Cárdenas-Rodríguez, N., & Mendoza-Torreblanca, J. G. (2024). Effect of Levetiracetam on Oxidant-Antioxidant Activity during Long-Term Temporal Lobe Epilepsy in Rats. International Journal of Molecular Sciences, 25(17), 9313. https://doi.org/10.3390/ijms25179313

Reviewer 2 Report

Comments and Suggestions for Authors

Thank you for submitting your manuscript to Brain Sciences. The idea is sound; however, here are  my comments

  • A graphical abstract is highly recommended
  • The introduction section is too short and needs to be increased. A figure explaining the mechanism of action of CLO is highly recommended to be added.
  • The study design and results sections are incomplete. This is the worst section, as measuring GSSH and malondialdehyde is insufficient to draw this conclusion. I would recommend performing the SeaHorse technique on the mitochondria to explore its function, also adding another ROS marker, as SOD is recommended 
  • Immunohistochemistry using any ROS marker is always recommended, where the staining of the affected section will be more obvious, instead of the Black and white micrographs you showed in your results 
  • Where is the study limitation section? Please add it at the end of the conclusion section 
  • in the materials and methods section. It should be rewritten to allow for reproducibility; also, a figure can help. What happened to the mice from day 35 (end of CLO treatment) to day 55 (sacrifice day), or from day 35 to day 85 (sacrifice day)
  • You based your study on refs 5 and 6, that depressive behavior was shown after chronic use of CLO, but have you measured any depressive behavior in those mice on day 85?
  • English needs to be revised as a lot of typos were detected
  • Reference section, the best section, as no self-citation was detected   

Comments on the Quality of English Language

  • English needs to be revised, as a lot of typos were detected  

Author Response

Please check the enclosed response coverletter.

Reviewer 3 Report

Comments and Suggestions for Authors The paper by Labra-Ruíz et al. is devoted to evaluating markers of oxidative stress and cellular damage in the prefrontal cortex and limbic system of Wistar rats exposed neonatally to long-term clomipramine (CLO) treatment. The authors found that early exposure to CLO can cause long-lasting neurochemical and structural alterations in examined brain regions. The paper is well structured, the 5 figures complement the text. Abstract gives the necessary information about the contents of the paper.The reference list covers the relevant literature (the authors cite 43 references).   However, I have some remarks: 1. In the Results section, please add the Student's t-test values. 2. Please explain why the hippocampus and amygdala were considered together and not separately. This should be added to the Methods section. 3. Please explain how you determined the sex of the rats on postnatal day 5 and what happened to the females? Were they removed from their mothers on day 5 or after day 25 along with their male brothers?

Author Response

Reviewer 3

The paper by Labra-Ruíz et al. is devoted to evaluating markers of oxidative stress and cellular damage in the prefrontal cortex and limbic system of Wistar rats exposed neonatally to long-term clomipramine (CLO) treatment. The authors found that early exposure to CLO can cause long-lasting neurochemical and structural alterations in the examined brain regions. The paper is well structured; the 5 figures complement the text. The abstract gives the necessary information about the contents of the paper. The reference list covers the relevant literature (the authors cite 43 references). However, I have some remarks:

First, we would like to thank the reviewer for his/her kind comments. Below, we address each of the points with the hope that the answers are satisfactory to you.

  1. In the Results section, please add the student's t-test values.

A= In accordance with your request, we have added the Student's t-test values. 

  1. Please explain why the hippocampus and amygdala were considered together and not separately. This should be added to the Methods section.

A= Histologically, the structures comprising the limbic system are easily identifiable, so histopathological analysis allowed for the differentiation of the hippocampus and amygdala. However, during the dissection process, the hippocampus was easily distinguishable, unlike the amygdala, whose boundaries were not clearly defined. Furthermore, the amount of tissue was minimal, which forced us to obtain the entire limbic structure for the determination of biochemical markers (lines 111--112).

  1. Please explain how you determined the sex of the rats on postnatal day 5 and what happened to the females. Were they removed from their mothers on day 5 or after day 25 along with their male brothers?

A= The sex of offspring can be determined from day 1 of birth by observing the distance between the anus and the genitals. At approximately 6–8 days of age, the nipples become visible in females, and from day 9 onward, the testicles become discernible when the abdomen is gently pressed. Sexing was performed on day 5 and confirmed on day 9. The female subjects remained with their respective mothers until weaning but did not receive any treatment. The male subjects were administered either clomipramine or a saline solution from day 5 to day 35. On the 21st day, the subjects were separated from their respective mothers and sisters. Notably, these subjects were utilized for breeding purposes or for any other work that required only females. 

Reviewer 4 Report

Comments and Suggestions for Authors

Comments 1. The quality of the introductory part can be significantly improved. Specify the mechanisms of cardiotoxicity, neurotoxicity of tricyclic antidepressants and clopyramine, namely due to the activation of active forms of oxygen 2. Materials and methods of research. Provide indicators of acute toxicity of clopyramine for Crimea and mice with different routes of administration. Provide a recalculation of the dose for children. Specify the method of administration to children (syringe and needles, volumes of administration). Specify in detail anesthesia and euthanasia 3. Discussion. This part is poorly written. There is no clear line for discussing the identified delayed mechanism of neurotoxicity. The intrigue remains until the end of the narrative. Who or what leads to the activation of oxidative stress and disruption of the histostructure of the brain after acute administration of clopyramine. There are two versions of events. The first. This is the formation of persistent mitochondrial dysfunction. The authors are developing this version. But very sluggishly. It is necessary to define mitochondrial dysfunction and the mechanisms of its formation. The authors had to determine glutathione in the mitochondria to confirm mitochondrial dysfunction. Determine other markers. Describe the consequences of myochondrial dysfunction for the brain. There is a second version of the activation of active forms of oxygen when taking tricyclic antidepressants. This is an increased level of serotonin, norepinephrine, a change in their transport systems. The authors do not discuss this version. There are no figures representing the authors' versions

Author Response

Reviewer 4

Comments:

  1. The quality of the introductory part can be significantly improved. Specify the mechanisms of cardiotoxicity and neurotoxicity of tricyclic antidepressants and clopyramine, namely due to the activation of active forms of oxygen.

A= In accordance with your request, we expanded the introduction section and included the cardiotoxicity mechanism of tricyclic antidepressants and clomipramine due to the activation of active forms of oxygen (lines 58--67).

  1. Materials and methods of research. Provide indicators of acute toxicity of clopyramine for Crimea and mice with different routes of administration. Provide a recalculation of the dose for children. Specify the method of administration to children (syringe and needles, volumes of administration). Specify in detail anesthesia and euthanasia.

A= a) Clopyramine (Chloropyramine; Chloropyramine) is an antihistamine used in allergic conditions. In this study, we used clomipramine (CLO), a high-affinity lipophilic tricyclic antidepressant. Regarding the "Crimean mouse" strain, we were unable to identify toxicity studies using this strain in the scientific literature. What a brief and systematic review has yielded is that the term "Crimean mouse" is associated with studies of the Crimean-Congo hemorrhagic fever virus in genetically modified mouse models.

Toxicity studies for antidepressant drugs have used mice, rats, rabbits, and dogs as study models. The toxicity analysis of CLO, specifically performed in mice, is based on mouse strains commonly used in biomedical research (BALB/C, C57BL/6, and NMRI), which demonstrate genotoxic effects and liver damage at doses ranging from 9.75 to 32.5 mg/kg.

The rat strains employed in the toxicity studies include Sprague-Dawley, Wistar, and WKY. The oral LD50in rats has been set at 613 mg/kg (Sigma-Aldrich, 2024), significantly higher, by an order of magnitude, than the doses used for therapeutic studies and those that have been shown to cause genotoxic and liver damage. For example, the highest therapeutic doses reviewed (50 mg/kg) represent less than 10% of LD50. This data, which is referenced in a safety data sheet, provides a reference point for the acute toxicity of clomipramine in the rat model (Sigma-Aldrich, 2024). Due to the lipophilic characteristic of CLO, the routes of administration are oral, subcutaneous, intramuscular, or intravenous. In general, most serotonin uptake inhibitor drugs have a prolonged-release design, so most studies refer to subchronic or chronic toxicity. Acute toxicity is associated with excessive and voluntary dose intake by the patient. The toxic dose is 10 times higher than the therapeutic dose, and this administration calculation depends on weight, physiology in the case of animal models, and concomitant conditions in humans. Regarding the indicators of acute toxicity by CLO, the severity of signs and symptoms varies depending on factors such as the amount of drug absorbed, the age of the patient, and the time elapsed since ingestion.

Critical manifestations of overdose include cardiac arrhythmias, severe hypotension, seizures, and CNS depression, including coma and death. These correspond to an intake greater than 300 mg/day. 

  1. b) This is not a clinical study, so there are no proposals for new doses or routes of administration of clomipramine in the pediatric population. However, the information about the administration of CLO in children indicates that it is only indicated for the treatment of nocturnal enuresis from the age of 5 years (if organ failure has been ruled out) and Compulsive Disorder (OCD). In the pediatric population, administration is oral via pills, tablets, or capsules. The initial dose is 10 mg/day, which increases over 10 days to 20 mg/day in children 5 to 7 years of age. For children 8 to 14 years of age, the dose gradually increases, according to medical management, to 50 mg/day. In adolescents 14 years of age and older, and depending on medical management and the pathology being treated, the maximum dose may reach 100 mg/day. Intravenous and intramuscular administration are strictly NOT authorized in children. 

REFERENCES

  1. -Asociación Española de Pediatría. Comité de Medicamentos. (2020). Clomipramina. Recuperado de https://www.aeped.es/comite-medicamentos/pediamecum/clomipramina
  2. -Agencia Española de Medicamentos y Productos Sanitarios (AEMPS). (s.f.). Ficha técnica de Anafranil.
  3. -Cayman Chemical. (2025). N-Desmethylclomipramine (hydrochloride) Safety Data Sheet.
  4. -Sigma-Aldrich. (2024). Clomipramine Hydrochloride Safety Data Sheet.
  5. -Toxbase. (s.f.). Clomipramina
  6. -Wilson M, Tripp J. Clomipramine. In: Stat Pearls. [internet] Treasure Island (FL): Stat Pearls Publishing;2025 Jan. Avaible from: http://www.nbci.nlm.nih.gov/books/NBK541006.
  1. c) In the present study, two methods of euthanasia were considered: exsanguination by barbiturate overdose (histological analyses) and decapitation (oxidative stress marker determination). Exsanguination was performed under deep anesthesia induced with xylazine (Rompum 10 mg/kg) and subsequently pentobarbital (120–210 mg/kg) injected into the peritoneum as part of the intracardiac perfusion procedure. After weaning, the mothers were administered an overdose of pentobarbital (120–210 mg/kg, i.p.). Conversely, decapitation was carried out without the administration of anesthesia since the use of any pharmaceutical agent could alter the biochemical parameters. The procedure was performed via a guillotine that had been meticulously sharpened, and the environments were sanitized with unscented soap and water, with the aim of mitigating stress caused by olfactory stimuli.
  1. This part is poorly written. There is no clear line for discussing the identified delayed mechanism of neurotoxicity. The intrigue remains until the end of the narrative. Who or what leads to the activation of oxidative stress and disruption of the histostructure of the brain after acute administration of clopyramine? There are two versions of events. The first. This is the formation of persistent mitochondrial dysfunction. The authors are developing this version. But very sluggishly. It is necessary to define mitochondrial dysfunction and the mechanisms of its formation. The authors had to determine glutathione in the mitochondria to confirm mitochondrial dysfunction. Determine other markers. Describe the consequences of mitochondrial dysfunction for the brain. There is a second version of the activation of active forms of oxygen when taking tricyclic antidepressants. This is an increased level of serotonin and norepinephrine and a change in their transport systems. The authors do not discuss this version.

A= In accordance with your request, the discussion section included more information related to the active forms of oxygen induced by CLO and its role in mitochondrial dysfunction, which could suggest that CLO induces oxidative stress, modifications in neurotransmitter systems, and brain structurechanges in our rat model (lines 366--388, 403--416, and 453--472).

  1. There are no figures representing the authors' versions.

A= In accordance with your suggestion, we included a figure of the CLO's known mechanism of action and the one proposed by us at the end of the discussion.

Round 2

Reviewer 1 Report

Comments and Suggestions for Authors

After first round of the manuscript revision,  authors  have taken into account all comments. I recommend this manuscript for publication in Brain Sciences.

Author Response

We greatly appreciate your valuable comments and the time you took to review our manuscript.

Sincerely,

Dr. Daniel Santamaria del Angel.

Reviewer 2 Report

Comments and Suggestions for Authors

Thank you for addressing the comments 

While I highly recommend to perform immunohistochemistry on the tissue and do not rely on ref 5 and 6 without checking the rat behavior, I admit that the article is now way better.

Author Response

Reviewer 2

While I highly recommend to perform immunohistochemistry on the tissue and do not rely on ref 5 and 6 without checking the rat behavior, I admit that the article is now way better:

A= Again, we acknowledge and appreciate your suggestions; however, lacking additional tissue, it is impossible for us to do what you requested. However, given the nature of this brief communication, we believe that the findings are valuable enough to be shared. We are currently contemplating the use of immunohistochemical assays in conjunction with the assessment of reactive oxygen species (ROS), superoxide dismutase (SOD), catalase (CAT), glutathione peroxidase activity, nitrosative stress, and behavioral analysis for future studies. Furthermore, in response to your comments, we have added a timeline of the experimental design to improve the description of the methods and improved the figure caption of CLO mechanism of action known and proposed by us.

Respectfully,

Dr. Daniel Santamaría del Ángel

Reviewer 4 Report

Comments and Suggestions for Authors

The authors have added the necessary information to the manuscript. The article "  Clomipramine induced oxidative stress and morphological alterations in the prefrontal cortex and limbic system of neonatal rats" can be published.

Author Response

(The authors gave the same response as above.)
